# Formulation of Topical Dosage Forms Containing Synthetic and Natural Anti-Inflammatory Agents for the Treatment of Rheumatoid Arthritis

**DOI:** 10.3390/molecules26010024

**Published:** 2020-12-23

**Authors:** Tunde Jurca, Liza Józsa, Ramona Suciu, Annamaria Pallag, Eleonora Marian, Ildikó Bácskay, Mariana Mureșan, Roxana Liana Stan, Mariana Cevei, Felicia Cioară, Laura Vicaș, Pálma Fehér

**Affiliations:** 1Department of Pharmacy, Faculty of Medicine and Pharmacy, University of Oradea, 1st December Square 10, 410068 Oradea, Romania; jurcatunde@yahoo.com (T.J.); annamariapallag@gmail.com (A.P.); marian_eleonora@yahoo.com (E.M.); laura.vicas@gmail.com (L.V.); 2Department of Pharmaceutical Technology, Faculty of Pharmacy, University of Debrecen, Nagyerdei Körút 98, 4032 Debrecen, Hungary; jozsa.liza@euipar.unideb.hu (L.J.); bacskay.ildiko@pharm.unideb.hu (I.B.); 3Doctoral School of Pharmaceutical Sciences, University of Debrecen, Nagyerdei Körút 98, 4032 Debrecen, Hungary; 4Institute of Healthcare Industry, University of Debrecen, Nagyerdei Körút 98, 4032 Debrecen, Hungary; 5Department of Psychoneuroscience and Rehabilitation Department, Faculty of Medicine and Pharmacy, University of Oradea, 1st December Square 10, 410068 Oradea, Romania; dr_ramona_suciu@yahoo.com (R.S.); cevei_mariana@yahoo.com (M.C.); felicia_cioara@yahoo.com (F.C.); 6Department of Preclinical Disciplines, Faculty of Medicine and Pharmacy, University of Oradea, 1st December Square 10, 410068 Oradea, Romania; marianamur2002@yahoo.com; 7Department of Pharmaceutical Biochemistry and Clinical Laboratory, Faculty of Pharmacy, University of Medicine and Pharmacy “Iuliu-Hațieganu” Victor Babeș Street, 400000 Cluj-Napoca, Romania; roxanaluc@yahoo.com

**Keywords:** rheumatoid arthritis, diclofenac sodium, *Calendula officinalis* extract, methyl salicylate, anti-inflammatory agent, antioxidant, surfactants, topical formulation

## Abstract

Topical anti-inflammatory and analgesic effect for the treatment of rheumatoid arthritis is of major interest because of their fewer side effects compared to oral therapy. The purpose of this study was to prepare different types of topical formulations (ointments and gels) containing synthetic and natural anti-inflammatory agents with different excipients (e.g.,: surfactants, gel-forming) for the treatment of rheumatoid arthritis. The combination of Non-Steroidal Anti-Inflammatory Drugs (NSAIDs), diclofenac sodium, a topical analgesic agent methyl salicylate, and a lyophilized extract of *Calendula officinalis* with antioxidant effect were used in our formulations. The aim was to select the appropriate excipients and dosage form for the formulation in order to enhance the diffusion of active substances and to certify the antioxidant, analgesic, and anti-inflammatory effects of these formulations. To characterize the physicochemical properties of the formulations, rheological studies, and texture profile analysis were carried out. Membrane diffusion and permeability studies were performed with Franz-diffusion method. The therapeutic properties of the formulations have been proven by an antioxidant assay and a randomized prospective study that was carried out on 115 patients with rheumatoid arthritis. The results showed that the treatment with the gel containing diclofenac sodium, methyl salicylate, and lyophilized *Calendula officinalis* as active ingredients, 2-propenoic acid homopolymer (Synthalen K) as gel-forming excipient, distilled water, triethanolamine, and glycerol had a beneficial analgesic and local anti-inflammatory effect.

## 1. Introduction

Rheumatism affects the joints and their peripheral structure (tendons, ligaments, muscles). It takes various forms depending on its location, symptoms, severity, or the age of the patient [1]. Rheumatoid arthritis is an inflammatory type of arthritis that can destroy the joint cartilage and cause deformity resulting in irreversible long term disability. This disease often affects the shoulder and knees [2,3]. The therapy of rheumatoid arthritis includes the administration of NSAIDs, systemic, and local corticotherapy, or physiotherapy [4].

The inflammatory process of rheumatism is determined by the secretion of proinflammatory cytokines such as tumor necrosis factor (TNF-α) and interleukin (IL-1) into the synovial cavity. Their release results in increasing vascular cellular infiltration, increasing production of protein C by hepatocytes, increasing osteoclast activity with bone erosions. These defensive reactions produce progressive tissue damage resulting in joint injuries, functional disability, and pain that is decreasing quality of life [5]. Pharmacotherapy includes non-steroidal anti-inflammatory drugs (e.g., diclofenac, ketoprofen, and naproxen) for modulating/reducing the inflammatory process by the inhibition of the production of prostaglandins in the body. The production of ‘harmful’ prostaglandins is prevented by inhibiting the inducible cyclooxygenase (COX-2) enzyme [6,7].

Topical non-steroidal anti-inflammatory drugs (NSAIDs) is a mainstay treatment for joint pain and provide at least equivalent analgesia, improvement in physical function, and reduction of stiffness compared with oral NSAIDs but have fewer systemic side effects [8].

Methyl salicylate (Salicylic acid methyl ester) is a volatile organic compound, the salicylic acid methyl ester, and is synthesized by several plant species [9,10]. It is a rubefacient that improves blood circulation and causes topical analgesic effect [10]. There are several topical formulations containing diclofenac sodium with methyl salicylate for the treatment of rheumatoid arthritis. These medications reduce pain, swelling, and joint stiffness, and helps to improve the ability to move and flex the joint [11].

There is a well-recognized connection between oxidative stress and chronic inflammation in rheumatism [12]. Reactive Oxygen Species (ROS) have been implicated as mediators of tissue damage in patients suffering from rheumatism. The turnover of the oxidant/antioxidant balance may lead to tissue damage. Therefore, the co-administration of antioxidants is also important in rheumatic disease [13,14].

Various topical pharmaceutical formulations contain *Calendula officinalis* extract due to its complex composition, as it is rich in many biologically active substances like flavonoids, carotenoids, glycosides, and steroids. It also contains polycarbohydrates, which play a role in wound healing and in the regulation of cellular permeability [15,16,17]. Carotenoids have been shown to have anti-inflammatory properties, while flavonoids have antioxidant and antimicrobial activities [17]. The main flavonoid component of *Calendula officinalis* is quercetin that has antioxidant effects by scavenging reactive oxygen and nitrogen species, but also targets prominent proinflammatory signaling pathways, such as STAT1, NFκB, and MAPK [18]. The extract has significant anti-inflammatory activity by the inhibition of the production of proinflammatory cytokines, COX-2 enzyme, and prostaglandin synthesis [16,19,20].

Physicochemical, pharmacokinetic, and pharmacological parameters are important in the qualitative assessment of topical products in order to obtain treatment compliance [21,22]. The concentration of active substances in joint tissues is also important [23]. The rheological behavior, the drug release form, and the topical dosage form determine the skin penetration of active substances. However, these parameters often depend on the type of topical dosage form and the excipients [22].

Different types of surfactants (sucrose ester SP 70 and Empicol LZ/N) were used for the formulation of ointments. Surfactants in topical preparations increase skin permeability, affect the physicochemical properties of the formulation, but may cause skin irritation [24]. Sucrose esters are natural and biodegradable excipients with penetration-enhancing, emulsifying, and solubilizing behaviors without side effects [25]. Empicol LZ/N is an anionic surfactant, which is the dried powder of sodium lauryl sulfate. This excipient is widely used in topical pharmaceuticals as an emulsifying agent. However, it may cause changes in the stratum corneum properties: transepidermal water loss and erythema may occur due to the application [26,27].

Different types of Carbomer gel-forming excipients (Synthalen K, Carbopol 974P and Pemulen TR1) were used for the formulation of gels. Carbomer is a synthetic high-molecular weight polymer of acrylic acid that helps to formulate low irritancy topical dosage forms, providing optimal aesthetics, feel, and drug penetration [28].

Pemulen TR1 is a polymeric emulsifier with high molecular weight copolymers of acrylic acid and C10-C30 alkyl acrylate crosslinked with allyl pentaerythritol. Pemulen polymer excipients contain both hydrophilic and hydrophobic portions and serve as excellent gel-forming polymers. Synthalen K is also a Carbomer type, synthetic, 2-propenoic acid homopolymer excipient, widely used in topical dosage forms due to its safety and versatility. It provides excellent stability at high viscosity and produces elastic and thick formulations for gels [28,29].

The main objectives of the present study were to formulate different types of topical dosage form containing *Calendula officinalis*, diclofenac sodium, and methyl salicylate and evaluate its therapeutic efficacy in rheumatoid arthritis. The aim of the study was to achieve a synergistic anti-inflammatory effect of the three active substances and to choose the best topical dosage form and excipients in order to achieve a good diffusion profile with analgesic and anti-inflammatory effects.

Rheological characterization and texture analysis of formulations were performed to evaluate the pharmaco-technical properties of formulations. In vitro membrane diffusion study was used to determine the release of active substances (diclofenac sodium, quercetin form *Calendula officinalis* extract and methyl salicylate). Diffusion coefficients and release rates were also calculated. MTT cytotoxicity assay was performed on HaCaT cells in order to prove the safety profile of our topical formulations. Free radical scavenging of the formulations was measured to demonstrate the beneficial antioxidant effect of the formulations. VAS (Visual analog scale) was used for pain assessment, and the thickness of synovium was also measured to determine the decrease of inflammation of patients suffering from rheumatoid arthritis.

## 2. Results

### 2.1. Determination of Macroscopic Properties of Topical Formulations

First of all, an assessment of the macroscopic characteristics of the formulations prepared was performed immediately after preparation. Thus, all ointments (Formula A–C) prepared had a homogeneous, orange appearance, while gels (Formula D–F) were translucent. The organoleptic properties of the preparations did not change after three months stored at room temperature. As a result, our formulations met the official requirements for visual, olfactory, and tactile features [30,31].

### 2.2. Rheological Characterization of Ointments and Gels

The determinations of the rheological characteristics have been performed to evaluate the pharmaco-technical properties and to assess which formulation is best suited for the application on the skin as the viscosity might affect the spreadability of the semi-solid formulations. To describe the rheological behavior of the preparations, we used the method by which the variation of viscosity was determined depending on the shear rate, in a time interval.

Knowing the effects of the formulation variables on its physical and mechanical properties is important because these effects can influence ointment technology, use (ease of application on the skin and bioadhesion), and therapeutic activity (bioavailability of incorporated drug substances).

Figure 1 presents the results of the rheological characterization of the six formulations with APIs.

The viscosity test provides information on the flow behavior of semi-solid formulations, by determining the flow curves (reograms) and the viscosity of the systems. These properties specific to each type of ointment and gel bases (lipogel, O/W emulsion base or hydrogel) can be modified in the presence of various substances in the composition, which perform different roles (emulsifier-Empicol LZ/N or sucrose ester SP 70, cosolvent-glycerin, humectant-absorption promoter-Lanolin).

According to the rheological experiments, the viscosity of the ointments (Formula A–C) and gels (formula D–F) depends on the magnitude and time of the mechanical effect (shear rate). Increasing shear stress was observed in all cases due to increasing shear rate.

Significant thixotropic and pseudoplastic behavior was detected in all formulations except Formula A. This meant that the semi-solid structure collapsed under mechanical action (shear) and became a fluid-like system. That was the reason for the appearance of the hysteresis effect (Figure 1a–f). It can be seen that each formulation was structurally viscous or shear thinning material, as the viscosity decreased with increasing shear rate in all cases.

As showed in Figure 1a, Formula A had the highest viscosity, which could be explained by the hydrocarbon type ointment base. The up curves and down curves practically coincided in this case. Therefore, a significant hysteresis effect was not detected under the given experimental conditions. Less thixotropic behavior of this formulation reflected the ability to rebuild its initial structure more rapidly after the removal of shear force [32]. Overall, lower viscosity values were observed for the gels than for the ointments, as might be expected. The lowest value was measured for Formula D (Figure 1d).

### 2.3. Texture Analysis

The penetrometry test was evaluated with the formulations with or without the active ingredients. For the analysis of the texture profile, a Brookfield CT3 Texture Analyzer was used. The probe was set to penetrate the sample containers at a depth of 25.0 mm and a speed of 2.0 mm/s. The recordings for the force exerted on the probe were made using Texture Pro Software. The determinations were performed in triplicate at 24.5 ± 0.5 °C. Figure 2 shows the compression force in Newton (N) needed for the probe to penetrate into the formulations in comparison with the control samples. The control samples were the same composition without diclofenac sodium, *Calendula officinalis* extract, and methyl salicylate.

According to our measurements, different compositions need different amounts of compression force. The resistance of gels (Formula D–F) showed significantly lower value compared to the ointment formulations (Formula A–C). According to the results of the compression test gels are more appropriate compositions for the sake of applicability. The highest compression force value was measured in the case of Formula A (140.33 ± 5.13 N in the case of composition with APIs and 146.00 ± 7.54 N without APIs). This hard composition structure may hinder the liberation of the active ingredients. Significant (*p* < 0.05) differences were not detected between Formula B and C, D and E, D and F, and E and F. The addition of APIs did not statistically change the compression force values in any of the cases (Table 1).

### 2.4. pH Measurement

The pH was determined potentiometrically, using a portable digital pH meter. Table 2 shows the pH values of the formulations. During the preparation of the ointments and gels, our goal was to prepare formulations with a pH close to the natural skin surface pH, which is around 4.7 [33]. As for gel preparation, adjusting the proper pH is very important in order to initiate gel formation. The most favorable pH range for this is around 6.5–7.0 [34]. Unneutralized polymers (Synthalen K, Carbopol, Pemulen) usually have a pH range of 2.5–3.5 depending on the polymer concentration. These unneutralized dispersions have very low viscosities, so the addition of triethanolamine is needed to increase pH [35]. Optimal viscosity can be achieved in pH ranges of 5.5–7.0 [36].

### 2.5. Permeation Studies

In vitro permeation profiles of Formula A–F were investigated with Franz diffusion method. Figure 3, Figure 4 and Figure 5 show the results of the in vitro permeation profiles of diclofenac sodium, methyl salicylate and quercetin, respectively, across isopropyl myristate (IPM) impregnated cellulose acetate membrane. Six independent Franz cells filled with ethyl alcohol (30%) were used during the experiments. Samples were taken from the receptor phase after 15, 30, 60, 90, and 120 min. The permeation profiles of the active ingredients showed dependence on the type of ointment/gel base used.

#### 2.5.1. Permeation Profile of Diclofenac Sodium

The average cumulative percentage of diclofenac sodium that penetrated through the membrane (%) was plotted against time (minute) in Figure 3.

Comparing the drug penetration profiles from ointments and gels, it can be concluded that the diclofenac sodium was present in higher amounts in the receptor phase in the case of gels. Formula D proved to be the most appropriate formulation according to the diffusion study. In this case, the cumulative amount of diclofenac sodium permeated after 2 h was 79.62 ± 0.91% (7.96 ± 0.09 mg). Diclofenac sodium permeation profiles from Formula E and F were very similar. At the end of the examination, the total diffused drug amount was 66.86 ± 2.31% (6.69 ± 0.23 mg) in the case of Formula E and 67.19 ± 3.76% (6.72 ± 0.38 mg) for Formula E.

According to our experiments, ointments demonstrated lower diffusion values compared with the gels. From formulation B, the amount of diclofenac sodium permeated through the membrane was 29.90 ± 0.56 % (2.99 ± 0.05 mg), from Formula C, it was 33.57 ± 1.08% (3.36 ± 0.11 mg) and from Formula A, it was the smallest amount (24.93 ± 1.56%, 2.49 ± 0.16 mg) after 2 h and the diffusion rate was also slower.

In order to evaluate the kinetics of diclofenac release from the ointments and gels, the fit of the results obtained in the in vitro permeation studies were analyzed by zero order mathematical models. Diclofenac sodium release rate (k) was determined from the slope of the amount of drug permeated per unit area and the square root of time. The diffusion coefficient (D) of the drug was calculated from the amount of drug released per unit area, the initial concentration, and the diffusion time [37,38]. D and k values are presented in Table 3.

Ordinary one-way ANOVA and Tukey’s multiple comparison tests were performed to compare diffusion coefficient values of formulations with each other. Significant differences are marked with asterisks in Table 4. Significant differences (*p* < 0.05) were not detected between the diffusion coefficient values of Formula A and B, B and C, E and F. In the other cases, the difference is significant.

#### 2.5.2. Permeation Profile of Methyl Salicylate

Permeated amount of the methyl salicylate was also measured. The percentage of methyl salicylate that permeated through the cellulose acetate membrane was plotted against time (minute) in Figure 4.

According to the results, methyl salicylate permeation was better in the case of the gels. The fastest and highest drug permeation was observed for Formula D, similar to the diclofenac sodium diffusion. The total permeated amount of methyl salicylate was 53.37 ± 1.86% in the case of the Synthalen K polymer containing gel composition, which was measured after 2 h diffusion time. Carbopol 974P (Formula E) and Pemulen TR1 (Formula F) containing gels were able to approach this value, as the maximum amount of active substance diffused was 49.91 ± 0.78% and 49.72 ± 1.48%, respectively. For Formula B and C ointments, it was found that there was no significant increase in the amount of drug permeated after 30 min. The worst membrane penetration occurred for the ointment composition containing Vaseline and Lanoline (20.30 ± 0.49%).

The release rate (k) of methyl salicylate was also calculated from the slope of the amount of released methyl salicylate per unit area versus the square root of time. The diffusion coefficient (D) of the API was estimated from the initial concentration, the amount of drug released per unit area, and the diffusion time. Release rates and diffusion coefficient values are listed in Table 5 [37,38].

The highest release rate and diffusion coefficient were detected in the case of Formula D. Ordinary one-way ANOVA and Tukey’s multiple comparison tests were performed to compare diffusion coefficient values of formulations with each other. Significant differences are marked with asterisks in Table 6. Between the diffusion coefficients of Formula B and C, E, and F, no significant differences were observed. In the other cases, significant differences (*p* < 0.05) were detected.

#### 2.5.3. Permeation Profile of Quercetin

Figure 5 shows the cumulative amount of quercetin permeated through the membrane from the different formulations (%) against time (minute). Each composition contained 5 g of *C. officinalis* extract, which is equivalent to 133.75 ± 2.8 mg quercetin [39].

Our results showed that the quercetin permeation was better from gel compositions; however, the difference between the gels and ointments was not as much as in diclofenac sodium diffusion study. Formula B (32.72 ± 0.69%) and C (33.60 ± 0.90%) were able to approach the diffusion profile of the gels. According to the results, quercetin release from emulsion type ointments were significantly higher than the release from Vaseline and Lanoline-based ointment (13.82 ± 1.06%).

The most appropriate composition regarded to the results of the diffusion studies was also the Formula D, which contained the active ingredients in a gel matrix formulated with the help of Synthalen K. In this case the cumulative amount of diffused quercetin was 45.01 ± 0.91% (12.04 ± 2.24 mg) after 2 h of diffusion time, for Formula E it was 40.42 ± 3.40% and for Formula F it was 37.25 ± 3.83%.

The release of quercetin from the different formulations after 2 h can be sorted based on the following descending order: D ˃ E ˃ F ˃ C ˃ B ˃ A

The release rate (k) of quercetin was calculated from the slope of the amount of quercetin released per unit area versus the square root of time. The diffusion coefficient (D) of the drug was estimated from the initial concentration of drug, the amount of drug released per unit area, and the diffusion time. Release rates and diffusion coefficient values are listed in Table 7 [37,38].

Ordinary one-way ANOVA and Tukey’s tests were also performed to compare diffusion coefficient values related to quercetin of formulations with each other (Table 8). Significant differences (*p* < 0.05) were not detected between the diffusion coefficient values of Formula B and C. In the other cases, the difference is significant.

### 2.6. Antioxidant Capacity

The 2,2-diphenyl-1-picrylhydrazyl (DPPH) method was used to test the ability of compounds to act as free radical scavengers or hydrogen donors, and to evaluate antioxidant capacity using a modified method of Brand Williams et al. [40]. The measurement of the color change (from dark violet to light yellow) correlated with the antioxidant capacity was performed at 517 nm on a Shimadzu UV-VIS spectrophotometer.

According to the DPPH assay the free radical scavenging activity of *C. officinalis* extract proved to be 65.34 ± 2.10% (Table 9). Radical scavenging activity (% inhibited reactive oxygen species (ROS)) was calculated for each composition with or without APIs. Significant differences were detected in every case between the compositions with and without APIs. In comparison, the antioxidant activity of the compositions containing *C. officinalis* extract was significantly higher than the activity of the same formulations but without the extract. Formulations without APIs did not show significant radical scavenging activity.

The results showed similar radical scavenging activity for Formula B and C. This is probably due to the fact that the ointment base is also very similar in these two cases. According to the DPPH assay, antioxidant activities of Formula E and F gels were also very similar. Among the formulations, only Formula D achieved 50% free radical scavenging activity, so it proved to be the most effective formulation. The lowest activity was measured for Formula A, which was due to an unfavorable ointment base (Vaseline and Lanolin) in terms of drug release.

### 2.7. In Vitro Cell Viability Study

The 2-(4,5-dimethyl-2-thiazolyl)-3,5-diphenyl-2H-tetrazolium bromide (MTT) cytotoxicity assays were performed on human keratinocyte cell line (HaCaT cell line) to examine the potential cytotoxic effect of the formulations. Samples were prepared with the Franz diffusion cells using phosphate buffered saline (PBS) as receptor solution. Formulations with the same compositions but without APIs were also tested. The cytotoxicity of *C. officinalis* extract dissolved in PBS was also examined in a concentration of 1.0 mg/mL and 0.5 mg/mL.

The results of the cytotoxicity investigation are shown in Figure 6. In the case of compositions without APIs, there was no significant difference in cytotoxicity compared to the same ointments and gels with APIs. Among ointments, Formula A demonstrated the highest cell viability value (88.92 ± 5.97%), while treatment with Formula B resulted in the largest decrease in cell viability, in this case, it was only 68.43 ± 5.62%, which was probably due to the Empicol LZ/N type emulgent in this formulation. The cytotoxicity of the emulgents depends on their structure: nonionic ones are considered safer than anionic, such as Empicol LZ/N. Of all formulations, Formula D proved to be the least toxic: the viability of the cells which were treated with this formulation resulted in 93.61 ± 1.49%. Among the gels, Formula E treatment showed the lowest cell viability (74.73 ± 2.23%). According to our results, it was found that Formula D proved to be a well-tolerated gel, showed the smallest decrease in the viability of the HaCaT cells. Based on our results, the viability of cells treated with C. officinalis extract dissolved in PBS (1.0 mg/mL and 0.5 mg/mL) were 84.02 ± 2.49% and 94.16 ± 1.48%, respectively.

### 2.8. The Anti-Inflammatory and Painkiller Effect on the Synovium

The anti-inflammatory potential of the selected ointment was based on a prospective study. After exclusion of patients (*n* = 115), a total of 50 patients were selected who accomplished the criteria for inclusion and completed the informed consent. Of these patients predominant were females 34 (68%) and 16 (32%) were male. The average age of females was 56.11 (±7.81) years and for the males was 52.7(±8.66) years. In this study 36 patients were diagnosed with rheumatoid arthritis of the knee, and 14 patients with rheumatoid arthritis of the shoulder (Figure 7).

In order to evaluate the therapeutic activity, Formula D was chosen because of a good display capacity and a constant release profile of diclofenac immediately after application. The anti-inflammatory effect of the preparation was obtained from the combination of diclofenac sodium, methyl salicylate and *C. officinalis* extract. Five grams of the preparations (Formula D with APIs or without APIs) were used for each application.

The local treatment with Formula D containing APIs in the experimental group was compared with a placebo treatment in a control group for 2 weeks. According to our experiments, there was a statistically significant decrease in pain between the first and the 14th day of treatment based on the Visual Analogue Scale (VAS). In the case of the patient whose knee was affected the average value at baseline was 6.06 ± 1.21 for the active group and 6.18 for the placebo group out of 10.0 (Figure 8). On the 14th day of treatment, it decreased to 2.36 ± 0.89 in the case of the experimental group, while it remained at 5.48 ± 0.54 for the placebo group. The difference between these two values was proved to be significant according to the unpaired t test (*p* ˂ 0.0001). The percentage change from baseline to the 14th day was 61.1 % for the experimental group. For shoulder pain, the percentage pain reduction from baseline was 47% in the active group. The difference between the experimental and the placebo group on the 14th day was also statistically significant (*p* < 0.0001).

In patients with arthritis, the thickening of the lining layer of the synovium may occur due to the inflammation [41]. The normal synovial membrane is a maximum of 1.8 mm thick, a value greater than this indicates synovial hypertrophy [42].

Ultrasound was used to evaluate the effectiveness of the therapy; the thickness of the synovial membrane (synovial hypertrophy) was measured ultrasonographically in the active and the placebo groups. In this regard the obtained results showed that the thickness of the synovium was also reduced significantly from the initiation in the case of patients with knee RA 3.22 ± 1.14 to 1.32 ± 0.91 in the active group on the 14th day (*p* ˂ 0.0001). For patients with shoulder RA the initial synovium thickness was reduced from 3.41 ± 1.39 to 1.92 ± 1.07 (Figure 9). The difference between the active and the placebo group on the 14th day was also statistically significant for RA of the knee (*p* < 0.0001). Synovial membrane thickness of the placebo groups did not change significantly during the trial. According to the measurement, the applied local therapy proved to be more effective in the case of RA of the knee, as the reduction of the thickness was 59% in the active group in comparison to 44% for the group with RA of the shoulder.

## 3. Discussion

The development of topical formulation for the treatment of rheumatoid arthritis without systematic side effects with local anti-inflammatory activity is highly anticipated.

In the present study, six different topical formulations containing the combination of diclofenac sodium, methyl salicylate, and *Calendula officinalis* extract were prepared. According to our previous experiments, the gel formulation with Synthalen K showed the best diffusion results and the lowest viscosity. As Bolla et al. described, the type of vehicle used may influence the bioavailability of the active ingredients in a topical dosage form. Viscosity is one of the most important factors for topical formulations as it may influence the release of drug by modulating the diffusion rate from the vehicles. In the case of the gels the higher diffusion rate could be due to their low viscosity, which resulted in enhanced drug release from these formulations [43]. Moreover, gels are also spread easily, so they are the best for treating large areas.

Cytotoxicity investigations were also performed on HaCaT cell line in order to prove the safety of the formulations. HaCaT cell line was derived from a man suffering from melanoma and has been proposed as an in vitro model for the study of the biocompatibility of topical formulations [44,45]. The MTT assay showed that Formula D was the least toxic among the formulations, as the viability of the cells was 93.61 ± 1.49% in this case.

According to our preformulation study data (rheological test, Texture analysis study, in vitro Franz diffusion test, MTT cytotoxicity test) and antioxidant test, Formula D (gel formulation) was selected for further in vivo examination.

The evaluation of the anti-inflammatory and analgesic effect of our formulation (Formula D) was performed by a prospective study in patients from the Rehabilitation Clinical Hospital from Băile Felix, Bihor, Romania. In our in vivo study, the age of participants was in a similar trend to those reported in the literature [46].

Pain is a symptom and as such the only person who can accurately appreciate its intensity is the person who feels it. In medical practice, it was necessary to objectify its intensity, so that it was proposed to use different assessment tools. Visual analog scale (VAS) is a method of pain assessment, being the most commonly used in clinical practice. The method is simple, non-invasive and easy to use by the patient, allows classification of pain into mild pain, moderate pain, and severe pain [47,48,49]. In a 3-week study performed by Efe et al., diclofenac gel was shown to be superior to placebo gel in relieving pain [50]. According to our study, the pain assessed by VAS SCALE was reduced significantly after 14 days of treatment with the gel formulation.

Keen et al. reported that the short-term synovial response can be detected ultrasonographically [51]. Before and after the treatment, the thickness of the synovium was evaluated ultrasonographically. The results showed that the treatment with the Formula D gel formulation had a beneficial effect on local inflammation and the thickness of the synovium reduced significantly compared to the placebo group.

The combination of *Calendula officinalis*, diclofenac sodium, and methyl salicylate in the presented form of gel can be used in rheumatoid arthritis with good results. Bodhankar et al. stated that calendula oil had an enhancing effect on the in vitro percutaneous absorption of diclofenac sodium [52]. The combination of diclofenac sodium with methyl salicylate in topical nanoemulsion improved the clinical efficacy in osteoarthritis of the knee compared to those formulations that contained diclofenac sodium alone, indicating the additive effect of methyl salicylate [11].

Several articles reported the anti-inflammatory and antioxidant effect of *Calendula officinalis* extract. Topical application of a 70% ethanol extract of the flower to mice at a dose of 1.2 mg/ear (corresponding to 4.16 mg crude drug) reduced croton oil-induced ear edema by 20% [16]. The ethanolic extract of *Calendula officinalis* presented anti-inflammatory activities and acted in a positive form on the inflammatory and proliferative phases of the healing process of cutaneous wounds of Wistar female rats [53].

*Calendula officinalis* can prevent oxidative stress, through the numerous polyphenols contained in its extract. Braga *et al*. stated that it neutralized reactive oxygen and nitrogen species (ROS and RNS) at concentrations as low as 0.2 μg/mL [54,55].

Several studies provide evidence for the involvement of ROS in the pathogenesis of rheumatoid arthritis and pointed to how adjuvant antioxidant therapy improves the course of the disease [14]. Van Vugt et al. described the clinical relevance of an antioxidant therapy and the beneficial effects of antioxidants on rheumatoid arthritis [13]. As it was described previously, Formula D showed the highest radical scavenging activity among the six formulations according to our antioxidant investigation.

Our work points out the differences between ointment and gel formulations with various excipients like surfactants and gel-forming polymers. The excipients influenced the rheological behavior, the cytotoxicity of formulations and the release of active substances, as well as the antioxidant effect. The favorable skin penetration and hence therapeutic efficacy, combined with a low potential for adverse effects on the skin, suggest that the gel formulation containing Synthalen K is a rational alternative to topical formulation for the treatment of rheumatologic conditions such as rheumatoid arthritis of the knee and the shoulder.

## 4. Materials and Methods

### 4.1. Materials

Empicol LZ/N was purchased from Innospec Performance Chemicals Italy SRL, CLW0025334 series. Diclofenac sodium (CAS Number: 15307-79-6), methyl salicylate (CAS Number: 119-36-8) and triethanolamine (CAS Number: 102-71-6) were purchased from Merck KgaA, Germany. Synthalen K was purchased from Elton Corporation SA Romania, lot 1518F65A. White Vaseline and Lanolin (Adeps lanae anhydicus) were purchased from Zhongbao Chemicals Co., Ltd., Pharm Grade USP37. Cera flava and glycerol were purchased from Nordische Oelwerke Walther Carroux GmbH & Co. KG, Hamburg, Germany.

Sucrose ester SP 70 [Sucrose stearate (CAS Number: 84066-95-5)] was a kind gift from Sisterna (Roosendaalc, The Netherlands). Cetostearyl alcohol [hexadecan-1-ol;octadecan-1-ol (CAS Number: 67762-27-0)] and isopropyl myristate [propan-2-yl tetradecanoate (CAS Number: 110-27-0)] were purchased from Hungaropharma Ltd., (Budapest, Hungary).

The MTT [2-(4,5-dimethyl-2-thiazolyl)-3,5-diphenyl-2H-tetrazolium bromide)] dye, Dulbecco’s Modified Eagle’s Medium (DMEM), phosphate buffered saline (PBS), trypsin from porcine, ethylene-diamine-tetra-acetic acid (EDTA), heat-inactivated fetal bovine serum (FBS), L-glutamine, 2,2-diphenyl-1-picrylhydrazyl (DPPH), absolute ethanol, ascorbic acid were purchased from Sigma-Aldrich (Budapest, Hungary). Nonessential amino acid solution and penicillin–streptomycin mix, GlutaMax™ supplement, 96-well plates, and cell culture flasks were obtained from Thermo-Fisher (Darmstadt, Germany). HaCaT cells (human keratinocyte cells) were obtained from Cell Lines Service (CLS, Heidelberg, Germany).

### 4.2. Preparation of Dry Calendula Officinalis Flower Extract

*Calendula officinalis* flowers were harvested from June to August 2019, from Bihor County, Romania. For the extraction of bioactive compounds from the plant a mixture of ethyl alcohol and distilled water was used as an extraction solvent. One hundred gram of *C. officinalis* flower sample was placed in a conical flask and 1500 mL of 96% ethanol solution were added. The mixture was extracted twice in an ultrasonic bath for 90 min at 45 °C [39]. The extracted dispersion was filtered through a cellulose membrane (0.45 µm pore diameter). The residue was repeatedly extracted with 60 % ethanol and water in the same conditions. After the extraction, the removal of the alcoholic fraction was done with centrifugation and the supernatant was evaporated to dryness in a rotary evaporator [56]. The dry *Calendula officinalis* flower extract was lyophilized (Christ Alpha 1-2 LD plus). The lyophilized powder was used for further studies.

The phytochemical analysis of *Calendula officinalis* (*Asteraceae* family) was carried out in our previous study using the HPLC method with a device equipped with a diode array detector [57] indicating the presence of polyphenols (*p*-coumaric acid, caffeic acid, gallic acid) and flavonoids (quercetin, epicatechin, rutine, myricetin). One kg of lyophilized *Calendula officinalis* was equivalent to 628.04 ± 0.15 mg quercetin. As the conditions of harvesting, extraction and lyophilization of the plant were the same in these experiments the concentration of quercetin (in the in vitro dissolution study) was determined according to our previous assay [57].

### 4.3. Preparation of Topical Formulations

#### 4.3.1. Preparation of Ointments

For the purpose of achieving efficient delivery systems, three ointment formulations have been prepared by incorporating diclofenac sodium, *Calendula officinalis* flower extract and methyl salicylate in different types of ointment bases.

Formulation A was prepared by hot blending of the two components, the Vaseline, and the Lanolin. To avoid overheating the components were added in reverse order to the melting point and melted on a Universal Thermostatic Water Bath.

Formulation B was prepared in two steps being an O/W emulsion ointment base. Thus, the lipophilic phase was obtained by melting the cetylstearyl alcohol and the cera flava (yellow beeswax) on the water bath. The aqueous phase formed from the aqueous solution of Empicol LZ/N and glycerin was added to the oily phase at 50°C ± 2°C with continuous stirring (DLS Stirrer, Velp Scientifica Germany, 500 rpm). We have chosen Empicol LZ/N as an anionic surfactant, extremely versatile, used for its excellent optimization properties for bioadhesive preparations.

Formulation C was also an emulsion ointment like Formulation B. The lipophilic phase was obtained by melting the cetylstearyl alcohol, glycerol and cera flava on the water bath and homogenized. The aqueous phase formed from the aqueous solution of sucrose ester SP 70 nonionic emulsifying agent and propylene glycol, which was also heated to 60 °C. The aqueous and lipophilic phases were homogenized and cooled down to 25 °C. The composition of the three ointment bases is presented in Table 10.

The final step was the addition of the active ingredients. Diclofenac sodium was used at a concentration of 1.00%, lyophilized extract of *Calendula officinalis* extract 5.00% and methyl salicylate 12.5%.

#### 4.3.2. Preparation of Gels

Gels are colloidal coherent systems of semi-solid dosage forms prepared from colloid-sized inorganic particles and organic macromolecules. Gels are usually clear or transparent and have high liquid content, they usually contain a large amount of water (about 85–90%) in which the concentration of gelling agent is 5–20%. In the gel formation process, the macromolecule absorbs water through swelling.

The compositions of formulated gels are shown in Table 11. Formula D was made by dispersing Synthalen K in a mixture of glycerin and water. Glycerol contributes to ensuring good viscosity, giving it a translucent appearance. Glycerol and distilled water were added to the vessel, the mixture was homogenized, and Synthalen K was gradually added under stirring. Stirring was continued with a mixer type stirrer for better dispersion. The mixture was allowed to stand for 2 h. When the Synthalen K hydration process has been completed, the basic component (formed by the dispersion of triethanolamine in distilled water) is added in small portions, and stirring is continued with the mixer until the gel is clarified.

In the case of Formula E and F, the appropriate amount of polymer was added to the specified amount of water and glycerol, and then the gelation process started with the addition of triethanolamine. Formula E contained Carbopol 974P polymer, while Formula F was prepared with Pemulen TR1 polymer. The active ingredients were then suspended in a given amount of the prepared gels.

### 4.4. Determination of Macroscopic Properties

Formulation of the ointments and gels was followed by evaluating their macroscopic characteristics. Organoleptic properties such as color, odor, physical appearance, and homogeneity were evaluated by visual perception immediately after preparation and after 3 months stored at room temperature (20 ± 1 °C) [30,31].

### 4.5. Rheological Analysis

In order to characterize the physicochemical properties of the formulations studied, we performed rheological measurements. The rheological properties were determined using RheolabQC Rotational Rheometer with Peltier heating system. Data were analyzed with RheoPlus Rheometer Software (32 V3.10 21003407-33024), the mean values were presented and the standard deviation (SD) was calculated. The viscosity curves of the formulations were determined by rotation tests at a controlled shear rate. 0.50 g of the formulations were applied to the cup of the concentric cylinder measuring system (CC27-SN11271, d = 26.7 mm) of the rheometer. The concentric cylinder measuring system (according to DIN EN ISO 3219 and DIN 53019) was considered to be appropriate for the examination of semi-solid dosage forms. Measurements were made at shear rate ranging from 2.0 to 50.0 s^−1^ (120 to 300 rpm) and then in a descending order at 24.0 °C [36].

### 4.6. Texture Analysis (Compression Force)

The quantification of textural properties was carried out using CT3 Texture Analyzer (Brookfield, Middleboro, MA, USA) equipped with TA5 Cylinder type probe (12.7 mm diameter, 35 mm length). Compression test was performed, and the hardness of formulations was measured.

A jar filled with the given formulation was positioned 5.0 cm under the probe of the device. The probe was lowered to the surface of the sample with a speed of 1.0 mm/s. After reaching the surface, the probe penetrated to a depth of 25.0 mm with a speed of 2.0 mm/s and the force exerted on the probe was recorded by Texture Pro CT Software (Brookfield Engineering Laboratories, MA, USA). The texture analysis was performed at room temperature (24.5 ± 0.5 °C). All measurements were done in triplicate. The means and the standard deviations were calculated.

### 4.7. Determination of pH

The pH was determined potentiometrically, according to Romanian Pharmacopoeia (FR X), using a portable digital pH meter (Sension™ 1, Hach Company, Loveland, CO, USA). Five-gram ointment/gel was added to 20 mL of distilled water previously heated to 37 ± 2°C and stirred vigorously for 1 min. After cooling, the dispersion was filtered, and the pH was determined in the filtrate. Each determination was made in triplicate.

### 4.8. In Vitro Permeation Studies

Franz diffusion system (Microette-Hanson system, model 57-6AS9, USA) with a diffusion area of 1767 cm^2^ and a volume of 6.5 mL for the receptor chamber was used in our study. Hydrophilic synthetic cellulose acetate membranes with 25 mm diameter and 0.45 μm pores were used for the in vitro test.

The receptor chamber in each diffusion cell was filled with 30% freshly prepared, heated and de-aerated ethanol. Synthetic cellulose acetate membranes were impregnated with isopropyl myristate for 30 min prior to use, then mounted between the Franz diffusion cell donor and acceptor compartment. A sample of 1.00 g was put on each membrane and placed on the top of the diffusion cell. The diffusion cells were tightly closed by fixing the dosing capsule with a clamp, thus preventing the vehicle evaporation and ensuring the integrity of the formulation throughout the study [11]. The system was maintained at 32 ± 1°C and the receptor medium was shaken continuously (350 rpm) by means of a magnetic stirrer to avoid the effects of the diffusion layer. One ml of the receptor solution was taken at 15, 30, 60, 90, and 120 min and replaced with a fresh receptor medium to maintain constant volume during the assays. The diclofenac sodium content of the samples was measured at 275 nm using a UV spectrophotometer (Shimadzu, Tokyo, Japan) [58]. The diffused amount of methyl salicylate was measured at 237 nm [59]. The specificity of the method was evaluated by recording the spectra of diclofenac sodium and methyl salicylate dissolved in the receptor phase (30% ethanol) in a concentration of 20 µg/mL between 200 nm to 400 nm. According to our measurements, the spectra of these active ingredients did not show overlapping at 237 nm. However, at 275 nm the methyl salicylate showed minimal absorbance, which was taken into consideration in the calculation of the concentration of diclofenac sodium. Both diclofenac sodium and methyl salicylate absorbances were measured also at 275 nm and 237 nm (the wavelengths of maximum absorbance of each substance) in concentrations of 5, 10, and 20 µg/mL. According to the validation data the absorbance which was measured at 275 nm composed of 94.6% diclofenac sodium and 5.4% methyl salicylate, so the value measured during the UV spectrophotometric method was decreased with 5.4% in the case of the diclofenac sodium.

During the measurement of methyl salicylate, due to its large amount, the samples were diluted 100-fold with the receptor medium. The main component of *Calendula officinalis* extract, the quercetin was measured at 370 nm [17]. As a blank sample, 30% alcohol was used. The calibration curves of diclofenac sodium, methyl salicylate and *Calendula officinalis* extract (quercetin) were determined before the spectroscopic measurements. For a good evaluation of the results, a control sample of an industrially manufactured pharmaceutical product containing 1% declared diclofenac sodium was used. Each ointment and gel formulation was tested in 6 replicates and data was presented as mean ± SD.

Diclofenac sodium, methyl salicylate, and quercetin release rate (*k*) was determined from the slope of the amount of drug released per unit area (µg/cm^2^) versus the square root of time (min^½^). The diffusion coefficient (*D*) of the drug was calculated from the drug concentration at a given t time (*Q*, µg/cm^2^), the initial concentration ($C_{0}^{\prime }$), and the diffusion time (*t*):(1)$$D=\frac{Q^{2}\times \;\pi }{{\left(2\left[C_{0}^{\prime }\right]\right)}^{2}\;\times \;t}$$

### 4.9. Antioxidant Capacity Test

DPPH free radical scavenging activity of the prepared ointment and gels was measured using a modified method of Brand Williams et al. [40]. Samples from formulations with or without APIs were collected using the Franz diffusion cells. Ointments and gels were examined with three individual cells where absolute ethanol was used as the receptor phase. Samples of 1 mL were taken at the 120th minute, when the diffusion of the active ingredients reached their maximum. The antioxidant activity of *C. officinalis* extract (15.0 mg/mL ethanol solution) was also investigated.

For the antioxidant capacity test 2.0 mL of DPPH radical solution (0.06 mM) in absolute ethanol was added to 900 µL of absolute ethanol. A total 100 µL of sample was added to the mixture of DPPH and absolute ethanol. The reaction mixtures were kept in the dark for 30 min to incubate. DPPH reacted with antioxidant compounds, which can donate hydrogen. When DPPH accepted hydrogen radical the reaction resulted in color change from dark violet to light yellow. The measurement of the absorbance was carried out with UV-spectrophotometer (Shimadzu Spectrophotometer, Tokyo, Japan) at 517 nm, with absolute ethanol as background [60]. Absorbance was used to calculate antioxidant activity percentage (inhibited ROS %) with the formula
AA% = 10 − {[(Abssample − Absblank) × 100]/Abscontrol}(2)
where Abs_sample_ was the absorbance of the mixture of the given sample and the DPPH solution, Abs_blank_ was the absorbance of the absolute ethanol and Abs_control_ was the absorbance of the mixtures of the negative control and the DPPH solution [61]. Alcoholic solution of ascorbic acid (0.25 mg/mL) was used as a standard in order to check the correctness of the measurement [62]. As negative control 2.0 mL of DPPH solution (0.06 mM) diluted with 1.0 mL absolute ethanol was applied.

### 4.10. Cell Culturing

The HaCaT cell line was used in cell viability assay. Cells were grown in a plastic cell culture flask (Nunc™ EasyFlask™, Thermo-Fisher, Darmstadt, Germany) in Dulbecco’s Modified Eagle’s Medium, supplemented with 10% (*v/v*) heat-inactivated fetal bovine serum (FBS), 4 mmol/L L-glutamine, 1% (*v/v*) non-essential amino acids solution, 100 IU/mL penicillin, and 100 μg/mL streptomycin at 37 °C in an atmosphere of 5% CO_2_ [63].

The culture medium was changed twice per week. The cells were routinely maintained by regular passaging. The cells used for the experiments were between passage numbers 20 and 40.

### 4.11. In Vitro Cell Viability Assay

For the cell viability assay test, solutions of the formulations were prepared with the Franz diffusion cells. Each formulation was examined with three individual diffusion cells where PBS was used as the receptor phase instead of alcohol. Samples of 1 mL were taken at the 120th minute, when the diffusion of the active substances reached their maximum.

Samples from *Calendula officinalis* extract were prepared by dissolving 1.0 mg and 0.5 mg lyophilized extract in 1.0–1.0 mL PBS. The components were mixed by magnetic stirrer at 25 °C for 2 h in each case.

The cytotoxic effects of the compositions were evaluated using the 3-(4,5-dimethylthiazol-2-yl)-2,5-diphenyl-tetrazolium bromide (MTT) test [64]. The MTT test is a colorimetric method based on the ability of the cells to metabolize. The activity of the NAD(P)H-dependent mitochondrial oxidoreductase enzyme is examined, which converts the yellow MTT dye into a water-insoluble purple formazan crystal [(E, Z)-5-(4,5-dimethylthiazol-2-yl)-1,3-diphenylformazane] [65,66].

HaCaT cells were seeded on 96-well plates at a final density of 10^3^ cells/well and allowed to grow in a CO_2_ incubator at 37 °C for 5 days. Then the medium was removed, test solution was added, and the cells were incubated for 3 h with the samples. After the removal of the samples, the cells were washed with 0.5 mL PBS, and 0.5 mg/mL MTT solution (MTT salt dissolved in PBS) was added to the wells. The cells were incubated for another 3 h. Finally, the MTT solution was removed and the purple formazan crystals were dissolved in acidic isopropanol (isopropanol:1.0 N hydrochloric acid = 25:1). The absorbance was measured at 570 nm against a 690 nm reference with FLUOstar OPTIMA Microplate Reader (BMG LABTECH, Offenburg, Germany). Cell viability was expressed as the percentage of the untreated control.

### 4.12. Randomised, Placebo-Controlled Clinical Trial

Ethical statement: All subjects gave their informed consent for inclusion before they participated in the study. The study was conducted in accordance with the Declaration of Helsinki, and the protocol was approved by the Ethics Committee of the Rehabilitation Clinical Hospital Băile Felix, Bihor, Romania, with the ethical number 12820/28.12.2018.

In order to evaluate the therapeutic activity, Formula D was chosen because this composition showed the best physicochemical properties and had the most appropriate release profiles of diclofenac sodium, methyl salicylate and quercetin immediately after application.

We carried out randomized prospective study on 115 patients with rheumatism abarticular (soft tissue), treated at the Rehabilitation Clinical Hospital Băile Felix, Bihor, Romania, between January and June 2019.

Inclusion criteria were as follows: patients older than 18 years of age with diagnosis of scapulo-humeralus periathritis or periathritis of the knee, or Quervain tendinitis or epicondylitis, patients who accepted the ultrasound investigation and who were diagnosed with tendonitis. Exclusion criteria were degenerative rheumatism, inflammatory rheumatism, dementia, cancer, infectious diseases and dermatological diseases.

Fifty patients were selected who have accomplished the criteria for inclusion and completed the informed consent. Among them 36 patients were diagnosed with RA of the knee, and 14 with RA of the shoulder.

Patients were randomized by a computer code to one of two treatment groups: Active group was treated with the active ingredients containing preparation. Placebo group was treated with the formulation containing no active ingredients. They underwent the treatment with the gel twice a day (morning and evening), for 14 days. Five grams of Formula D preparation were used in each application.

All the patients taken in the study completed a questionnaire to assess pain using the Visual Analogue Scale (VAS) at baseline and after 14 days of local topical treatment with ointment. On the VAS scale, patients were able to place their responses (pain intensity) anywhere on a 10-cm-long line with “no pain” (at level 0) and “unbearable pain” (at level 10) verbal descriptors at the endpoints [67].

As ultrasound can detect short-term synovial response [50], assessments of the knee and the shoulder was carried out both longitudinally and transversely by high frequency ultrasound (US) in B-mode (Philips HDI 5000) and total synovial membrane thickness was measured using grayscale on the day of entry into the study and 14 days later. Grayscale synovial hypertrophy was defined according to Outcome Measures in Rheumatology (OMERACT) definitions [68]. Inflammation was characterized by the swelling of synovial lining, which was associated with an increase of synovial fluid, and an increased perfusion of the lining [69,70].

The measurement of the synovial thickness was performed according to the protocol recommended by European League Against Rheumatism (EULAR) and a study performed by Jan et al. [71]. In both cases, the largest anteroposterior diameter in the US image was evaluated.

## 5. Conclusions

While the systematic therapy of NSAID causes substantial side effects, including gastritis and gastric ulcer disease, renal impairment, hypertension, and thrombotic events, the topical administration of NSAIDs offers the advantage of local, enhanced drug delivery to affected tissues with reduced incidence of systematic adverse effects [72]. In our present study, the combination of diclofenac sodium, methyl salicylate with a natural component *Calendula officinalis* extract in gel formulation showed good diffusion property of drugs that may provide high tissue concentration leading to better and longer pain relief and anti-inflammatory effect. The proper local treatment of rheumatoid arthritis may also help patients to prevent immediate surgical intervention and to enhance quality of life activities.

## Figures and Tables

**Figure 1 molecules-26-00024-f001:**
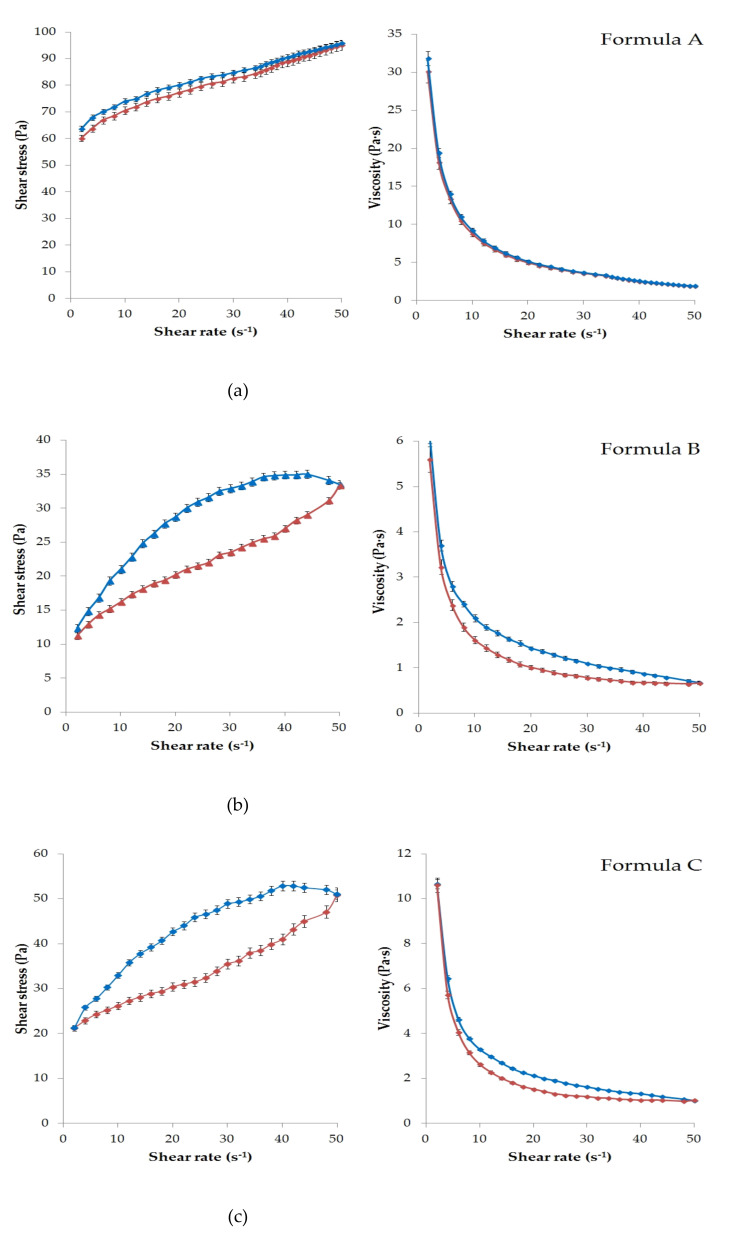
Flow curves of the ointment and gel samples plotted as shear stress (Pa) (left) and viscosity (Pa·s) (right) against shear rate. The rheological characteristics of (**a**) Formula A, (**b**) Formula B, (**c**) Formula C, (**d**) Formula D, (**e**) Formula E and (**f**) Formula F are presented. Each data point represents the mean ± S.D., *n* = 3. Shear rate was first increased from 0.01 to 50 s^−1^ (up curve, signed with blue) and then decreased from 50 to 0.01 s^−1^ (down curve, signed with red) to check possible hysteresis effects.

**Figure 2 molecules-26-00024-f002:**
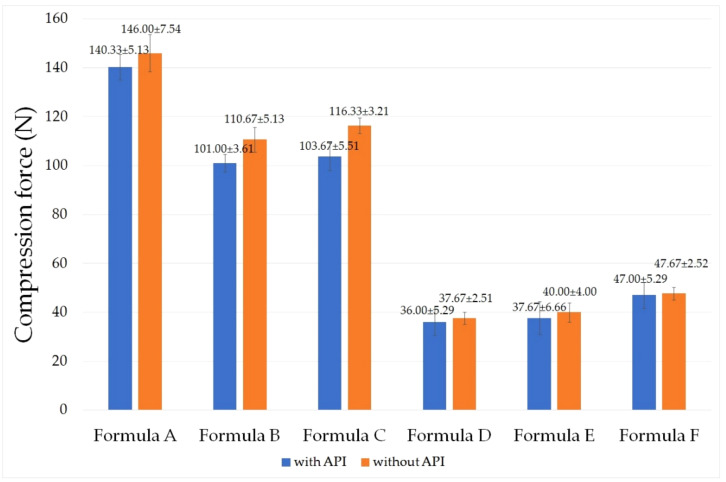
Result of the texture analysis of the formulations at 24.5 ± 0.5 °C, determined as compression force (N). Compositions containing the active pharmaceutical ingredients (APIs) were signed with blue columns, and compositions without APIs were signed with orange columns. Each data point represents the mean ± S.D., *n* = 3.

**Figure 3 molecules-26-00024-f003:**
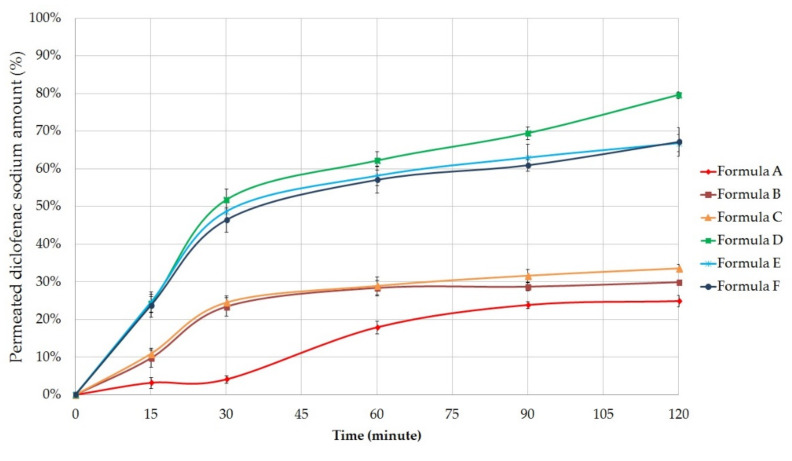
In vitro permeation profile of diclofenac sodium across isopropyl myristate (IPM) impregnated cellulose acetate membrane from the formulations. Bars represent mean ± S.D. (*n* = 6).

**Figure 4 molecules-26-00024-f004:**
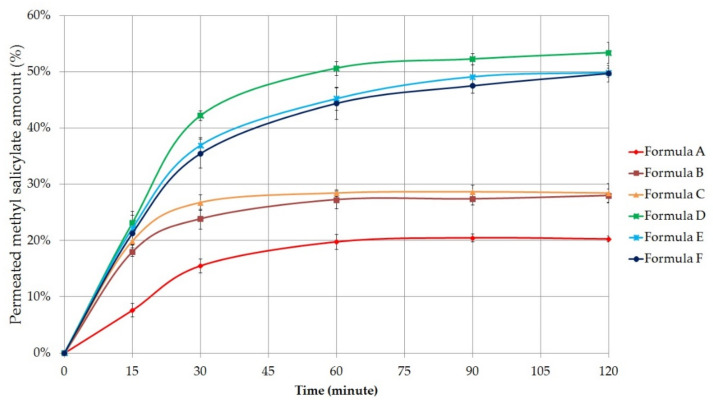
In vitro permeation profile of methyl salicylate across isopropyl myristate (IPM) impregnated cellulose acetate membrane from the formulation A–F. Bars represent mean ± S.D. (*n* = 6).

**Figure 5 molecules-26-00024-f005:**
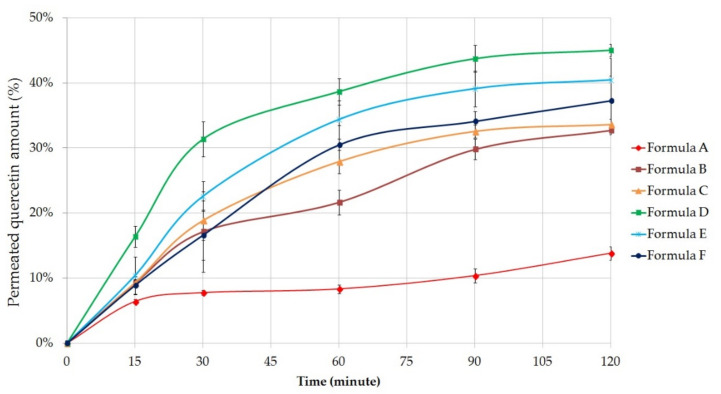
In vitro permeation profile of quercetin across isopropyl myristate (IPM) impregnated cellulose acetate membrane from Formula A–F. Bars represent mean ± S.D. (*n* = 6).

**Figure 6 molecules-26-00024-f006:**
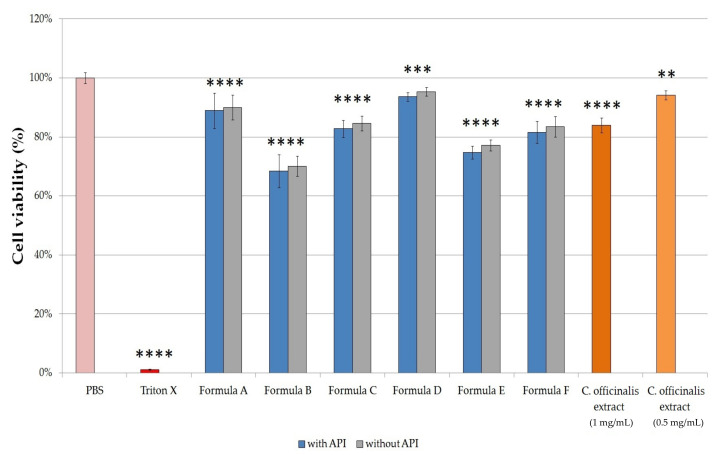
Cell viability assay (MTT assay) on HaCaT cells treated with Formula A-F and with *C. officinalis* extract dissolved in PBS in concentrations of 1.0 mg/mL and 0.50 mg/mL. Each data point represents the mean ± S.D., *n* = 10. Cell viability is expressed as the percentage of negative control (PBS), which was treated with PBS. The positive control was Triton X 100 (10% *w*/*v*). Ordinary one-way ANOVA with Dunnett’s multiple comparison tests were performed to compare the different formulations and extracts with PBS. **, *** and **** indicate statistically significant differences at *p* < 0.01, *p* < 0.001, and *p* < 0.0001.

**Figure 7 molecules-26-00024-f007:**
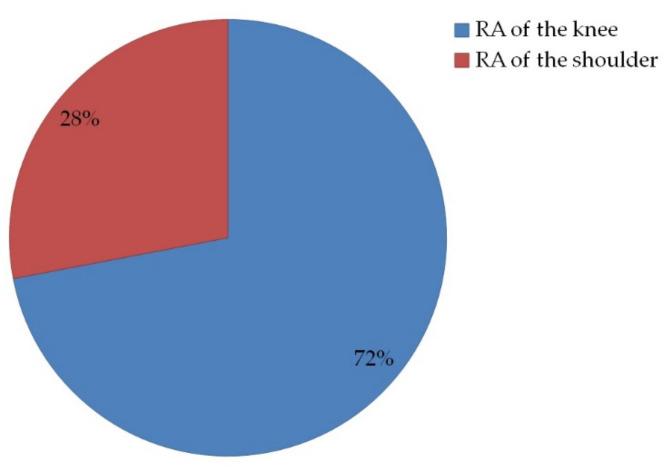
The distribution of rheumatoid arthritis (RA) diseases according to diagnosis. Thirty-six patients (72%) were diagnosed with rheumatoid arthritis of the knee, and 14 patients (28%) with rheumatoid arthritis of the shoulder.

**Figure 8 molecules-26-00024-f008:**
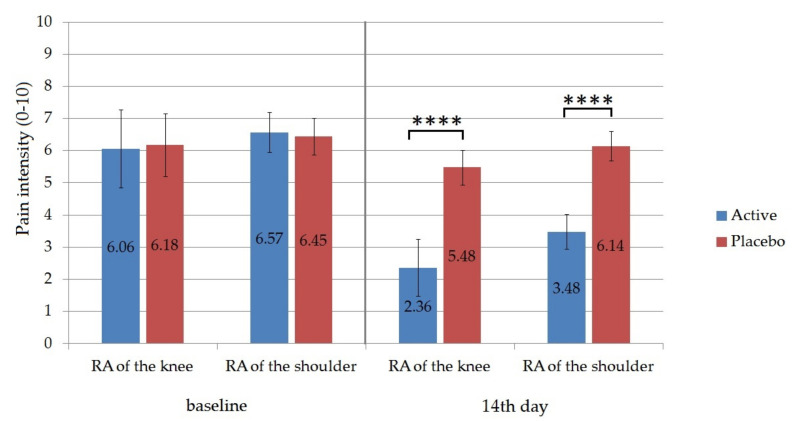
The evolution of the pain management (VAS scale scores). On the VAS scale, patients were able to place pain intensity anywhere on a 10-cm-long line with “no pain” (at level 0) and “unbearable pain” (at level 10) descriptors at the endpoints. Values represent means ± SD. Unpaired t tests were performed to compare the active and the placebo groups with each other. Significant differences are marked on the figure with asterisks (*p* < 0.0001). Active group was treated with Formula D containing APIs, while the placebo group was treated with the same formula without APIs. Significant differences were detected between the active and the placebo group on the 14th day of the clinical trial.

**Figure 9 molecules-26-00024-f009:**
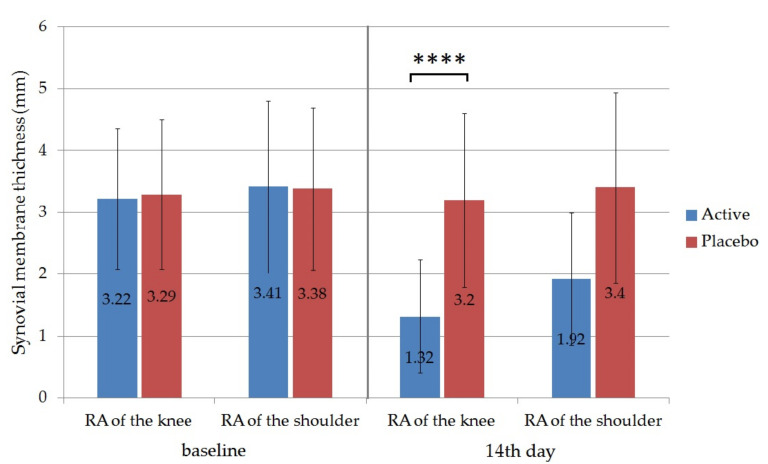
The ultrasonographically measurement of the synovial membrane thickness (mm). Values represent means ± SD. Unpaired t tests were performed to compare the active and the placebo groups with each other. Significant differences are marked on the figure with asterisks (*p* < 0.0001). Active group was treated with Formula D containing APIs, while placebo group was treated with the same formula without APIs. Significant differences were detected between the synovial thickness of active and the placebo group on the 14th day in the case of patients with RA of the knee.

**Table 1 molecules-26-00024-t001:** Results of Tukey’s multiple comparison test generated by the data of textural analysis.

Compared Data Sets	Level of Significance
Formula A c vs. Formula A	ns
Formula B c vs. Formula B	ns
Formula C c vs. Formula C	ns
Formula D c vs. Formula D	ns
Formula E c vs. Formula E	ns
Formula F c vs. Formula F	ns
Formula A vs. Formula B	****
Formula A vs. Formula C	****
Formula A vs. Formula D	****
Formula A vs. Formula E	****
Formula A vs. Formula F	****
Formula B vs. Formula C	ns
Formula B vs. Formula D	****
Formula B vs. Formula E	****
Formula B vs. Formula F	****
Formula C vs. Formula D	****
Formula C vs. Formula E	****
Formula C vs. Formula F	****
Formula D vs. Formula E	ns
Formula D vs. Formula F	ns
Formula E vs. Formula F	ns

c = control (formulations without APIs), ns = not significant, **** = *p* < 0.0001.

**Table 2 molecules-26-00024-t002:** The pH values of the compositions. Each data point represents the mean ± S.D., *n* = 3.

Composition	pH Value ± SD
Formula A	5.77 ± 0.05
Formula B	4.93 ± 0.03
Formula C	4.85 ± 0.04
Formula D	5.72 ± 0.02
Formula E	5.84 ± 0.04
Formula F	5.91 ± 0.02

**Table 3 molecules-26-00024-t003:** Diclofenac sodium release rate and the diffusion coefficient values related to the formulations A-F. Each data point represents the mean ± S.D., *n* = 6.

Composition	Release Rate	Diffusion Coefficient
	k·10^2^ (µg/cm^2^·h^1/2^) ± S.D.	D·10^5^ (cm^2^/min) ± S.D.
Formula A	114.98 ± 3.12	0.1302 ± 0.032
Formula B	128.07 ± 4.41	0.1873 ± 0.027
Formula C	141.13 ± 8.23	0.2361 ± 0.031
Formula D	324.44 ± 6.34	1.3282 ± 0.076
Formula E	277.78 ± 9.11	0.9365 ± 0.056
Formula F	276.27 ± 10.35	0.9459 ± 0.078

**Table 4 molecules-26-00024-t004:** Results of Tukey’s multiple comparison test generated by the data of diffusion coefficients.

Compared Data Sets	Level of Significance
Formula A vs. Formula B	ns
Formula A vs. Formula C	*
Formula A vs. Formula D	****
Formula A vs. Formula E	****
Formula A vs. Formula F	****
Formula B vs. Formula C	ns
Formula B vs. Formula D	****
Formula B vs. Formula E	****
Formula B vs. Formula F	****
Formula C vs. Formula D	****
Formula C vs. Formula E	****
Formula C vs. Formula F	****
Formula D vs. Formula E	****
Formula D vs. Formula F	****
Formula E vs. Formula F	ns

* = *p* < 0.05 and **** = *p* < 0.0001, ns = not significant.

**Table 5 molecules-26-00024-t005:** Methyl salicylate release rate and the diffusion coefficient values related to the formulations. Each data point represents the mean ± S.D., *n* = 6.

Composition	Release Rate	Diffusion Coefficient
	k (µg/cm^2^·h^1/2^) ± S.D.	D·10^5^ (cm^2^/min) ± S.D.
Formula A	1099.1 ± 32.96	13.491 ± 0.472
Formula B	1360.8 ± 40.82	25.626 ± 0.897
Formula C	1369.0 ± 41.07	26.491 ± 0.927
Formula D	2781.7 ± 82.45	93.255 ± 3.062
Formula E	2590.6 ± 77.72	81.557 ± 2.543
Formula F	2562.1 ± 76.86	80.919 ± 2.832

**Table 6 molecules-26-00024-t006:** Results of Tukey’s multiple comparison test generated by the data of methyl salicylate diffusion coefficients.

Compared Data Sets	Level of Significance
Formula A vs. Formula B	****
Formula A vs. Formula C	****
Formula A vs. Formula D	****
Formula A vs. Formula E	****
Formula A vs. Formula F	****
Formula B vs. Formula C	ns
Formula B vs. Formula D	****
Formula B vs. Formula E	****
Formula B vs. Formula F	****
Formula C vs. Formula D	****
Formula C vs. Formula E	****
Formula C vs. Formula F	****
Formula D vs. Formula E	***
Formula D vs. Formula F	****
Formula E vs. Formula F	ns

*** = *p* < 0.001 and **** = *p* < 0.0001, ns = not significant.

**Table 7 molecules-26-00024-t007:** Quercetin release rate and the diffusion coefficient values related to the formulations A-F. Each data point represents the mean ± S.D., *n* = 6.

Composition	Release Rate	Diffusion Coefficient
	k·10^2^ (µg/cm^2^·h^1/2^) ± S.D.	D·10^5^ (cm^2^/min) ± S.D.
Formula A	48.961 ± 2.43	0.0400 ± 0.004
Formula B	135.03 ± 3.98	0.2242 ± 0.021
Formula C	146.45 ± 5.01	0.2365 ± 0.032
Formula D	191.13 ± 5.87	0.4244 ± 0.039
Formula E	178,84 ± 4.32	0.3422 ± 0.018
Formula F	162.09 ± 5.98	0.2907 ± 0.024

**Table 8 molecules-26-00024-t008:** Results of Tukey’s multiple comparison test generated by the data of quercetin diffusion coefficients.

Compared Data Sets	Level of Significance
Formula A vs. Formula B	****
Formula A vs. Formula C	****
Formula A vs. Formula D	****
Formula A vs. Formula E	****
Formula A vs. Formula F	****
Formula B vs. Formula C	ns
Formula B vs. Formula D	****
Formula B vs. Formula E	****
Formula B vs. Formula F	**
Formula C vs. Formula D	****
Formula C vs. Formula E	****
Formula C vs. Formula F	*
Formula D vs. Formula E	****
Formula D vs. Formula F	****
Formula E vs. Formula F	*

* = *p* < 0.05, ** = *p* < 0.01 and **** = *p* < 0.0001, ns = not significant.

**Table 9 molecules-26-00024-t009:** In vitro free radical scavenging activity of the formulations with or without APIs and the antioxidant activity of *C. officinalis* extract (15 mg/mL). As positive control ascorbic (0.25 mg/mL) dissolved in ethanol (96%) was applied. The negative control was 2.0 mL of DPPH solution (0.06 mM) diluted with 1.0 mL absolute ethanol. Results presented as mean radical scavenging activity (inhibited ROS %) ± SD. Ordinary one-way ANOVA and Tukey’s multiple comparison tests were performed to compare compositions with or without APIs. Significant differences are marked with **** (*p* < 0.0001), showing the significance in the case of formulations containing APIs and the same formulation without APIs.

Composition	Radical Scavenging Activity (Inhibited ROS %)	
	with APIs	without APIs
Formula A	10.40 ± 3.32 ****	2.35 ± 0.41
Formula B	29.35 ± 1.34 ****	2.61 ± 1.55
Formula C	30.73 ± 1.21 ****	4.03 ± 1.67
Formula D	50.64 ± 1.47 ****	5.03 ± 2.01
Formula E	43.64 ± 2.14 ****	4.38 ± 2.52
Formula F	43.23 ± 1.16 ****	5.25 ± 1.96
*C. officinalis* extract (15.00 mg/mL)	65.34 ± 2.10	
positive control: Ascorbic acid (0.25 mg/mL)	83.07 ± 1.43	
negative control: DPPH solution + absolute ethanol	0.21 ± 0.12	

ROS = reactive oxygen species, APIs = active pharmaceutical ingredients, DPPH = 2,2-diphenyl-1-picrylhydrazyl.

**Table 10 molecules-26-00024-t010:** Qualitative and quantitative composition of ointments.

Composition	Formula A (g)	Formula B (g)	Formula C (g)
Distilled water	-	57.50	57.50
Lanolin	40.75	-	-
White Vaseline	40.75	-	-
Cetylstearyl alcohol	-	10.00	10.00
Cera flava	-	2.50	2.50
Empicol LZ/N	-	1.50	-
Sucrose ester SP70	-	-	1.50
Glycerol	-	10.00	10.00
*C. officinalis* extract	5.00	5.00	5.00
Diclofenac sodium	1.00	1.00	1.00
Methyl salicylate	12.50	12.50	12.50

**Table 11 molecules-26-00024-t011:** Qualitative and quantitative composition of gels.

Composition	Formula D (g)	Formula E (g)	Formula F (g)
Distilled water	70.40	70.40	70.40
Synthalen K	0.50	-	-
Carbopol 974P	-	0.50	-
Pemulen TR1	-	-	0.50
Triethanolamine	0.60	0.60	0.60
Glycerol	10.00	10.00	10.00
*C. officinalis* extract	5.00	5.00	5.00
Diclofenac sodium	1.00	1.00	1.00
Methyl salicylate	12.50	12.50	12.50

## Data Availability

The data presented in this study are available on request from the corresponding author. The data are not publicly available due to ethical reason.

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
