# Peer review of "Formulation of Topical Dosage Forms Containing Synthetic and Natural Anti-Inflammatory Agents for the Treatment of Rheumatoid Arthritis"

_molecules, 2020, doi:10.3390/molecules26010024_

Round 1

Reviewer 1 Report

The work is interesting but It should be revised in some points:

Abstract. It should be of easy understanding from the reader. Please specify Synthalen K and the composition of formulation D.

Introduction. Line 50. Please correct NSAIDS with NSAIDs.

Line 71. Specify ROS.

Line 75. Write calendula off. In italic. Check the manuscript.

Line 79. Write correctly the bibliography 17-19. Check the manuscript.

Line 175. Why the experiment was performed at 23°C? In method section It Is reported 24,5°C. 

Figure 2. Please write correctly y axes name.

Release studies. This part is not clear. Authors used a membrane impregnated with a lipophilic material. Generally this Is performed for a permeation study. Why authors speak about release, diffusion? In that cassa the paper membrane must be used. Please revise this part  ti be more clear.

Reference 9. Actually Europen Pharmacopoeia 10th Is used.

Author Response

Response to Reviewer 1

Dear Reviewer 1,

Thank you for reviewing the manuscript. We have revised our manuscript according to your suggestions.

Line 216 (Part 2.5.). This part of the study has been revised: “Release studies” has been corrected to “Permeation studies”, because the amount of the active substance was investigated that permeated across the cellulose acetate membrane. Parts 2.5.1., 2.5.2., 2.5.3. and 4.8. have been also revised and corrected in several places. The y axes of Figure 3-5 have been also changed from “Diffused… amount” to “Permeated… amount”.

Reference 9 has been corrected to European Pharmacopoeia 10th. Thank you for your comment.

Yours faithfully,

Pálma Fehér, PhD

Assistant professor

University of Debrecen, Faculty of Pharmacy

Department of Pharmaceutical Technology

Reviewer 2 Report

General comments: 

This manuscript describes the formulation and evaluation of topical formulations containing synthetic and natural anti-inflammatory agents for the treatment of rheumatoid (RA) arthritis.  The in vivo data presented in this manuscript adds value to this manuscript, in effect proving the value of topical treatment for RA.  The following comments are applicable to this manuscript: 

In several places the authors start sentences with numbers, for instance in line 695.  Rather write the number in words if it is the start of a sentence. 

Please refer to the attached manuscript as well in which smaller changes are indicated with “sticky notes” and applicable comments. 

Introduction

Line 50:   The authors refer to “general and local corticotherapy”.  What is meant with “general” therapy?  Does this mean “systemic” therapy?  If this is the case please use the word “systemic”. 

Line 74:   The authors state that Calendula officinalis extract is used in “various topical pharmaceutical formulations due to its complex composition”   The authors need to explain why the complex composition is a reason for using the extract in formulations. 

Line 106:   “The aim was to achieve a synergistic inflammatory effect..” change to  “The aim of the study was to achieve a synergistic anti-inflammatory effect…”. 

Results

Lines 119 – 123:  The authors present results on “Determination of macroscopic properties of topical formulations”  In the Materials and Methods section this section is not described.  Please add a section in the Materials and Methods section that briefly describes the applicable methods. 

Line 126:  The authors refers to “which formulation is best suited for surface display”  What does the authors mean with “best suited for surface display”?.  Please rephrase or describe this term. 

Line 176:  The authors refer to “creams”.  This should be ointments.  

Lines 179 – 187:  Rather than incorporating the “data labels” in the caption of Figure 2, I suggest to add them as data labels in the Figure itself.  Each “data label” should be assigned to the appropriate bar in the figure.    

Line 176:  The authors refer to the “natural skin surface pH”.  Please give the value.  

Lines 211 – 212:  Figure 2 is incorrectly identified as a diffusion profile.  It should be Figures 3, 4 and 5. Also state here that the diffusion results of methyl salicylate are also presented.  It is not mentioned here although the results are presented in Figure 4. 

Line 291:  The authors refer to the formulation with “the most appropriate composition”.  Most appropriate with regards to what parameter?  The authors need to state what is considered as appropriate as it is a relative term. 

Line 340 – 341:  The authors state that MTT assays were conducted to “examine the bioavailability of the formulations”.  Bioavailability cannot be measured with this test.  This test evaluates cytotoxicity.  This statement is factually incorrect.  Please correct. 

Line 349: The authors state that formulation B is the most cytotoxic.  Please try and offer an explanation for this. 

Lines 366 – 368:  This information is supplied in the Methods section.  This is duplicated information.  Consider removing it here. 

Materials and Methods

Line 514:  The authors fuse the term “solution”.  Use the term dispersion.  This is strictly speaking not a solution as it contained undissolved material that needed to be removed by filtration.  

Line 526:  Please supply the appropriate reference here as well.  Is it [56]?

Lines 614 – 616:      Please supply the validation results of the UV spectroscopic method.  Specificity needs to be proved as to indicate that the diclofenac and methyl salicylate did not interfere with each other's analysis.

Line 675:  Please change “…MTT salt solved…” to “…MTT salt dissolved….”

Line 690:  Please change “…criteria were the followings:…” to “…criteria were as follows:….”

Line 693:  Please change “…were the degenerative rheumatism,…” to “…were degenerative rheumatism,...”

Author Response

Response to Reviewer 2

Dear Reviewer 2,

Thank you for reviewing the manuscript. We have revised our manuscript according to your suggestions.

In line 383, 389, 520, 608, 660, 719 and 725 the number has been changed to words at the start of the sentence. In line 348 “The” has been inserted to the start of the sentence, before the number. In line 388 the number (5) was also changed to words.

In line 53 the word “general” has been corrected to “systemic”. In line 57 hepatocyte has been changed to hepatocytes.

In line 79 the importance of complex composition of C. officinalis extract has been described. The extract contains many pharmaceutically active ingredients (e.g. flavonoids, carotenoids and polycarbohydrates) which have different effects (e.g. antioxidant, wound healing, anti-inflammatory and antimicrobial activities). [Hadfield, R., Vlahovic, T., Khan, M.T. The Use of Marigold Therapy for Podiatric Skin Conditions. Foot Ankle J. 2008, 1 (7), 1., Ashwlayan, V.D.; Kumar, A.; Verma, M.; Garg, V.K.; Gupta, S. Therapeutic Potential of Calendula officinalis. Pharm. Pharmacol. Int. J. 2018, 6(2), 149‒155., Butnariu, M.; Coradini, C.Z. Evaluation of biologically active compounds from Calendula officinalis flowers using spectrophotometry. Chem. Cent. J. 2012, 6, 35.]

In line 114 “The aim was to achieve a synergistic inflammatory effect…” has been changed to “The aim of the study was to achieve a synergistic anti-inflammatory effect…”.

In line 116 “inflammatory” was corrected to “anti-inflammatory”. Sorry for our mistake.

A new section has been added to the Materials and Methods part as proposed, where the method of the Determination of macroscopic properties has been described. (Line 580)

In line 135 “…best suited for surface display” has been rephrased to “…best suited for the application on the skin…”.

In line 185 “creams” has been changed to “formulations”, because texture studies were conducted both with gels and ointments.

Results of the texture analysis have been added as data labels in the Figure 2 itself as proposed.

In line 208 the value of the natural skin pH has been supplied.

In line 218 Figure numbers have been corrected and methyl salicylate has been added to this sentence. Thank you for your comment and sorry for the mistake.

Line 242: The word “gels” has been added to the sentence.

Line 298: Formula D proved to be the most appropriate composition, because of its good results in the permeation study: the cumulative amount of diffused quercetin was the highest in this case. In the manuscript the given sentence has been changed to “The most appropriate composition regarded to the results of the diffusion studies

Line 348: “…examine the bioavailability of the formulations” has been corrected to “…examine the potential cytotoxic effect”.

Line 350: Franz diffusion cells were used only for the preparation of the samples. These samples then examined during the MTT assay.

Line 357: The largest decrease in the viability of the HaCaT cells was detected in the case of the Formula B treatment. It was probably due to the Empicol LZ/N emulgent in this formulation. The cytotoxicity of the emulgents depends on their structure: nonionic ones are considered safer than anionic, such as Empicol LZ/N. The “biocompatible excipients” phrase has been deleted from the section of “Preparation of ointments” (Line 536).

In the part 2.8. The anti-inflammatory and painkiller effect on the synovium the duplicated information has been deleted.

Line 388: This sentence has been rephrased. “…pharmacologically associated active ingredients” has been changed to “…diclofenac sodium, methyl salicylate and C. officinalis extract” in order to better understand.

Line 419: In case of RA of the shoulder the p value was 0.0588 in the measurement of the synovial membrane thickness, so the difference between the active and the placebo group was not significant statistically according to the unpaired t test.

Line 422: “the reduction of the thickness was 59% in the active group in comparison to 44% for the group with RA of the shoulder”. This sentence has been corrected as proposed.

Line 522: “Solution” has been changed to “dispersion” as proposed.

Line 534: The missing reference has been added. Thank you for your comment.

Line 542: “Universal Thermostatic Water Bath” has been corrected to singular form.

Line 628-638: The specificity of the UV spectrophotometric method has been supplied: The specificity of the method was evaluated by recording the spectra of diclofenac sodium and methyl salicylate dissolved in the receptor phase (30% ethanol) in a concentration of 20 µg/mL between 200 nm to 400 nm. According to our measurements the spectra and the absorbance maximum of these active ingredients did not show overlapping at 237 nm. However, at 275 nm the methyl salicylate showed minimal absorbance, which was taken into consideration in the calculation of the concentration of diclofenac sodium.

Both diclofenac sodium and methyl salicylate absorbance were measured also at 275 nm and 237 nm (the wavelengths of maximum absorbance of each substance).

Diclofenac sodium (20 µg/mL) absorbance

237 nm: 0.000

275 nm: 0.821

Methyl salicylate (20 µg/mL) absorbance

237 nm: 0.458

275 nm: 0.047

Diclofenac sodium (10 µg/mL) absorbance

237 nm: 0.000

275 nm: 0.409

Methyl salicylate (10 µg/mL) absorbance

237 nm: 0.228

275 nm: 0.023

Diclofenac sodium (5 µg/mL) absorbance

237 nm: 0.000

275 nm: 0.212

Methyl salicylate (5 µg/mL) absorbance

237 nm: 0.113

275 nm: 0.012

According to our measurements at 237 nm wavelength only the methyl salicylate had absorbance. However, at 275 nm the methyl salicylate showed minimal absorbance, which we took into consideration in the calculation of the concentration of diclofenac sodium. The absorbance which was measured at this wavelength composed of 94.6% diclofenac sodium and 5.4% methyl salicylate, so the value measured during the UV spectrophotometric method was decreased with 5.4% in the case of the diclofenac sodium (calculated with ratio-par).

Line 697: “MTT salt solved” has been changed to “MTT salt dissolved”.

Line 714: “…criteria were the followings…” has been changed to “…criteria were as follows...” as proposed.

Line 717: “…were the degenerative rheumatism,…” has been changed to “…were degenerative rheumatism,...” as proposed.

Yours faithfully,

Pálma Fehér, PhD

Assistant professor

University of Debrecen, Faculty of Pharmacy

Department of Pharmaceutical Technology

Round 2

Reviewer 2 Report

The authors have addressed my comments appropriately.